# Uncertainty Analysis for Image-Based Streamflow Measurement: The Influence of Ground Control Points

**Wen-Cheng Liu [1]** **, Wei-Che Huang [1] and Chih-Chieh Young [2,3,*]**

1   Department of Civil and Disaster Prevention Engineering, National United University, Miaoli 360023, Taiwan
2   Department of Marine Environmental Informatics, National Taiwan Ocean University, Keelung 20224, Taiwan
3   Center of Excellence for Ocean Engineering, National Taiwan Ocean University, Keelung 20224, Taiwan
*   Correspondence: youngjay@ntou.edu.tw; Tel.: +886-(2)-2462-2192 (ext. 6318)

**Abstract:** Large-scale particle image velocimetry (LSPIV) provides a cost-effective, rapid, and secure monitoring tool for streamflow measurements. However, surveys of ground control points (GCPs) might affect the camera parameters through the solution of collinearity equations and then impose uncertainty on the measurement results. In this paper, we explore and present an uncertainty analysis for image-based streamflow measurements with the main focus on the ground control points. The study area was Yufeng Creek, which is upstream of the Shimen Reservoir in Northern Taiwan. A monitoring system with dual cameras was set up on the platform of a gauge station to measure the surface velocity. To evaluate the feasibility and accuracy of image-based LSPIV, a comparison with the conventional measurement using a flow meter was conducted. Furthermore, the degree of uncertainty in LSPIV streamflow measurements influenced by the ground control points was quantified using Monte Carlo simulation (MCS). Different operations (with survey times from one to nine) and standard errors (30 mm, 10 mm, and 3 mm) during GCP measurements were considered. Overall, the impacts in the case of single GCP measurement are apparent, i.e., a shifted and wider confidence interval. This uncertainty can be alleviated if the coordinates of the control points are measured and averaged with three repetitions. In terms of the standard errors, the degrees of uncertainty (i.e., normalized confidence intervals) in the streamflow measurement were 20.7%, 12.8%, and 10.7%. Given a smaller SE in GCPs, less uncertain estimations of the river surface velocity and streamflow from LSPIV could be obtained.

**Keywords:** uncertainty analysis; image velocimetry; LSPIV; streamflow measurement; ground control points (GCPs); Monte Carlo simulation



## 1. Introduction

Streamflow measurements, which provide fundamental data regarding river discharge (by the measured velocity and bathymetry), play an essential role in hydro-environmental research, e.g., hydrological condition analysis, numerical model calibration and validation, water resource management, and hydraulic engineering planning and design [1–5]. However, the workflow for effective streamflow measurements can be quite tedious, time-consuming, difficult, and sometimes dangerous, especially during high-flow periods [6]. To gain the required hydrological information under different environmental constraints, continuous efforts to develop low-cost, safe, efficient, and accurate measurement techniques (less influenced by environmental or technical factors) have been made over the past decades [7,8].

In general, two types of approaches are used for streamflow measurement: direct or non-intrusive measurement. In the former group, several types of flow meters are directly placed into the river to measure the flow velocity. By considering the issue of safety, this method is not feasible for high-flow conditions. With advances in measuring instruments, acoustic Doppler current profilers (ADCPs) have been developed and applied to measure

bathymetry, velocity, and river discharge. Note that higher costs for the equipment and regular maintenance are required to use these instruments [9]. Moreover, there are some limitations. For example, the spatial variations in the flow field cannot be obtained if the instrument is fixed in one location. When an ADCP is mounted on a boat, poor weather conditions would hinder its measurement. Moreover, the operation of ADCPs in small streams would be difficult and unsuitable [10,11].

Alternatively, non-intrusive methods, with advantages in cost, safety, and efficiency, have drawn a lot of research attention (e.g., [12,13]). In particular, their non-intrusive nature can be used to solve the undesired inconveniences of conventional direct streamflow measurements, especially during flood events [11,14–17]. The basic idea of non-intrusive methods is to measure the surface velocity and then provide a reasonable estimation of discharge for various flow conditions. To facilitate surveys in the field, image-based large-scale particle image velocimetry (LSPIV), which efficiently measures the motions of floating objects (e.g., bubbles) in a rectangular grid, has been developed [18–33]. Without tracking particles, it can effectively analyze the surface velocity through the ripple pattern on the river surface. Additionally, stable results can be obtained even with low-resolution images (if there are no external interference factors). Based on LSPIV and successive images obtained from a fixed platform above riverbanks or mobile equipment, the variations in surface velocities in both space and time can be sufficiently resolved [12,34–36]. For more details and information on the characteristics of LSPIV, one can refer to the excellent work by Muste et al. [22]. Currently, the development of LSPIV measurement is growing and becoming popular. At present, a number of methods based on LSPIV for streamflow measurement are available, e.g., digital particle image velocimetry (PIVlab) [37], Kanada–Lucas–Tomasi image velocimetry (KLT-IV) [38], optical tracking velocimetry (OTV) [2], surface structure image velocimetry (SSIV) [39], and space–time image velocimetry (STIV) [3].

The application of LSPIV in streamflow measurement is promising, but there are still several concerns [40]. One of the most important issues is its accuracy and uncertainty resulting from camera intrinsic calibration (i.e., distortion adjustment for the camera) and extrinsic calibration (i.e., scaling, direction, and projection for a position in the physical domain). Thus, understanding the efficiency of LSPIV in achieving the goal of carrying out unbiased and less uncertain streamflow measurements is essential. There are various factors causing uncertainties when measuring river surface velocities with LSPIV. For example, uncertainties associated with environmental effects (light and shadow, wind speed, etc.) are difficult to quantify. Uncertainty due to the changes in camera parameters and the sizes of the interrogation area (IA) has been reported [41,42]. Note that ground control points (GCPs) are required in LSPIV measurements to obtain orthorectified images via projective transformation (i.e., the relation between the pixel and physical spaces). There are currently two ways to measure GCPs: (i) total stations (or electronic distance measuring devices) and (ii) the Global Positioning System (GPS). As the GPS is more expensive, GCPs are usually measured using total stations. In principle, the accuracy of LSPIV measurements can be improved when a sufficient number of GCPs are used [43]. The number of GCPs depends on how they are used. Generally, at least four to six GCPs should be included. Additionally, the surveys for GCPs should be conducted several times and averaged to obtain accurate results for the coordinates. It has been recommended that these crucial GCPs should be evenly distributed in the image and should not be coplanar [42]. Furthermore, the surveys of GCPs that could affect camera parameters through the solution procedure of collinearity equations might impose uncertainty on the streamflow measurement results [44,45]. To the best of our knowledge, information or rules in the literature are still quite limited for GCP quality control and its resultant effects.

In this study, the purpose is to explore the uncertainty in image-based streamflow measurements with the main focus on ground control points. The degree of uncertainty in LSPIV streamflow measurement was quantified using Monte Carlo simulation (MCS), in which a large number of camera parameters obtained from the collinearity equations and ground control points were randomly sampled under different standard errors. The study

area was Yufeng Creek, which is upstream of the Shimen Reservoir in Northern Taiwan. A monitoring system with dual cameras was set up on the platform of a gauge station to measure the surface velocity. To ensure the accuracy of image-based LSPIV, a comparison with conventional measurements using a flow meter was also conducted. Finally, a range of standard errors were introduced into the GCPs. The influence of GCPs on the surface velocity and discharge of the river is discussed and presented.

## 2. Study Site and Measuring Instruments

### 2.1. Description of Study Site

This study was conducted at Yufeng Creek in Yufeng Village, Jianshi Township, Hsinchu County, Northern Taiwan (Figure 1). The location of Yufeng Creek is in an upper catchment area of the Shimen Reservoir. Moreover, Yufeng Creek is an upstream tributary of the Danshui River, which flows through the metropolitan area of Taipei and New Taipei City. Therefore, streamflow measurement of the creek is quite important, as it provides necessary information for water resource management (water supply) and disaster mitigation (flood control).

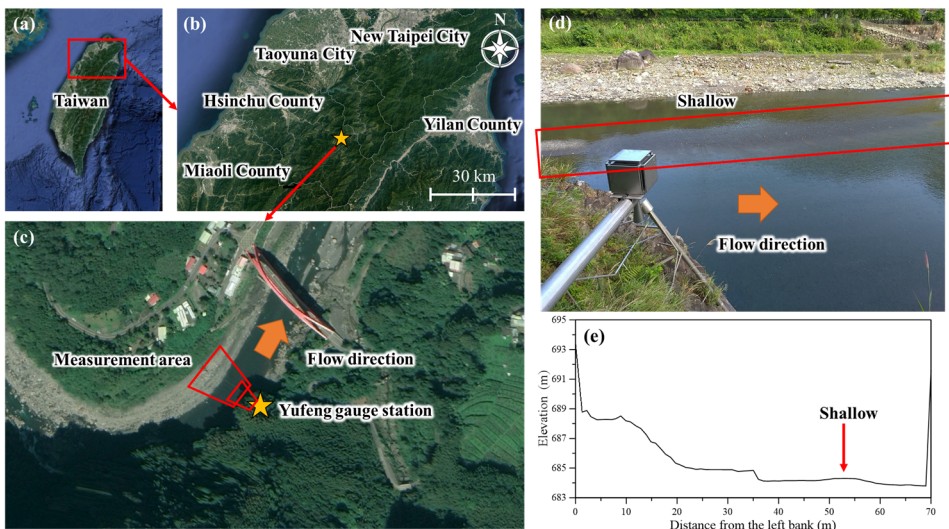

**Figure 1.** Location map of the study site. (**a**–**c**) study area; (**d**) image of study site; (**e**) river cross-sectional profile.

Yufeng Creek is about 75 m wide but is narrower (about 35 m) at the study site. The Yufeng water level gauge station is on the right bank of the creek (Figure 1c). During normal flow periods, elevation of the water level is about 684.5 m (above mean sea level) with a depth of 1.2 m and a maximum current speed of about 1.2 m/s. A shoal (see Figure 1d,e) that forms in the middle of the river would significantly affect the flow pattern. There are about 100 m-long groundsill works downstream of the Yufeng gauge station, which protects the river course without significant changes before and after flood events. The study site is suitable for long-term streamflow observations.

In this study, we considered four normal flow events for comparison of river surface velocity measured by LSPIV and the flow meter. The main reason for this was to ensure the safety of the surveyors who used the flow meter to measure the river flow for 10 h during the experimental periods. These benchmark data were utilized to evaluate the performance (accuracy) of LSPIV under its control point setup. Moreover, a representative event was used to demonstrate and discuss the uncertainty in LSPIV from the influence of control points. Note that the LSPIV system was shown to be capable of carrying out long-term continuous monitoring under severe weather conditions [16], although there was no typhoon event in this study.

### 2.2. Measuring Instruments

For digital image acquisition, a color industrial camera (i.e., model ICDA-acA1600-20gc) manufactured by the Germany Basler company (https://www.baslerweb.com/en/, accessed on 15 August 2022) was used (see Figure 2a). Its resolution is about two million effective pixels, and its maximum acquisition frequency is 20 fps (i.e., 20 frames per second), which fully complies with the image capture interval recommended by Gharahjeh and Aydin [46]. Note that the image matching analysis requires slight differences in two successive images. In a river, the ripples are quite dynamic. Therefore, this study used a high acquisition frequency (i.e., 20 fps) to capture the ripples and to avoid large distinctions in two images. A low-distortion lens (FV1520) produced by Myutron (https://www.myutron.com/en/lens/, accessed on 15 August 2022) was used along with the camera. The focal length was 15 mm, with a maximum distortion of about −0.09%. Other detailed specifications can be found on the manufacturers' official websites.

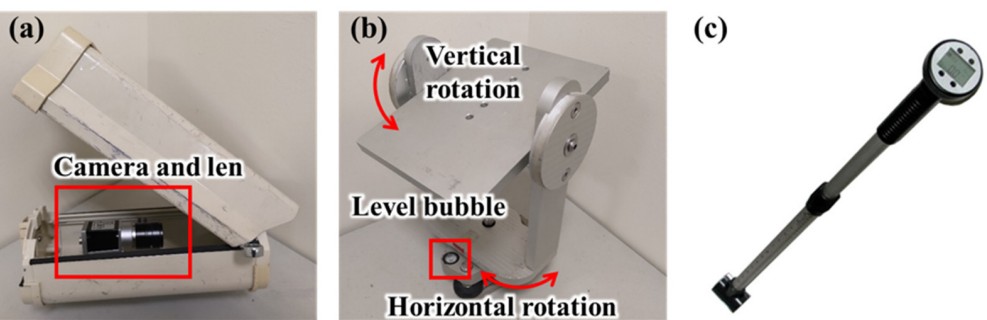

**Figure 2.** Instruments for measurement: (**a**) camera, lens, and protective shell; (**b**) swivel base; and (**c**) hand-held propeller digital flow meter.

The assembled camera and lens were placed into a protective shell (Figure 2a) that was then installed on a swivel base (Figure 2b). The swivel base can be rotated horizontally and vertically so that the camera's shooting range can be adjusted to the region of interest (ROI) for measurement. There was a level bubble on the swivel base to ensure a horizontal state. In other words, only the azimuth angle would be changed when the swivel base horizontally rotated. For the streamflow measurement, a system of dual cameras was set up on the platform of the Yufeng gauge station (691.6 m above mean sea level) to simultaneously photograph the water surfaces near the right and left banks of Yufeng Creek (see the red boxes in Figure 1c). The images from the dual cameras covering the entire cross-section of Yufeng Creek enabled measurements of surface velocity. Note that the resolutions of both images could be estimated by the image and object space coordinates of two adjacent pixels once the parameters of the collinearity equations are known. As oblique images were used, the farthest positions in both near-field and far-field images were taken to calculate the resolution. The farthest point of the near-field image was about 20 m from the right bank, with a resolution of 4.35 mm/pixel. For the far-field image, the farthest point was 35 m, and the resolution was 23.5 mm/pixel.

In order to offer a basis for comparison with the LSPIV measurements, an acoustic Doppler current profiler (ADCP) and a hand-held propeller digital flow meter were used to measure the river surface velocity in this study. However, the ADCP only provides river velocities at a depth below 20 cm, while its sensors (i.e., acoustic transmitter and receiver) must be placed under the water surface. On the other hand, the FP111 can measure near-surface velocities because the diameter of the propeller is 5 cm. Therefore, this study decided to utilize the FP111 for river surface velocity measurements. The hand-held propeller digital flow meter (FP111 type) manufactured by Global Water is shown in Figure 2c. The digital flow meter shares a similar principle to the Price flow meter, but it provides a convenient means for surface velocity measurement within a range from 0.1 m/s to 6.1 m/s, with an accuracy of ±0.1 m/s. Regarding its measurement method, the surveyor pulls a cross-sectional line and uses the FP111 to measure the surface velocities

every meter along the line. Finally, the mean surface velocity is computed, which can then be used to validate the results from LSPIV. The procedures for both measurements are illustrated in Figure 3.

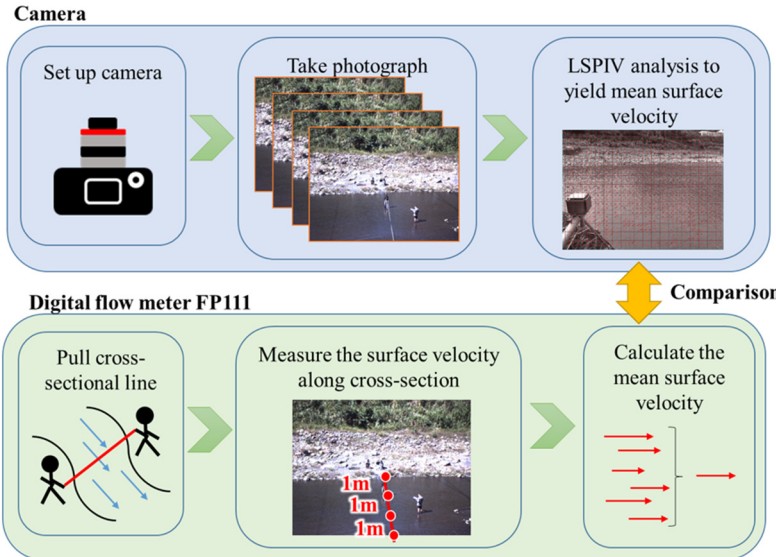

**Figure 3.** Measurement procedure for using image-based surface velocimetry and a propeller digital flow meter.

## 3. Methods: LSPIV Measurement and Uncertainty Assessment

### 3.1. LSPIV Measurement

To apply LSPIV for surface velocity analysis in river flows, the key procedures include the solution of collinearity equations and the matching of images. Furthermore, the cross-sectional mean velocity can be estimated from the surface velocity by a coefficient ($k$). Finally, the discharge is obtained by the product of the mean velocity and the cross-sectional area. The overall procedures are summarized as follows.

#### 3.1.1. Collinearity Equations

To analyze any point in the object space of an image, the so-called collinearity equations can be established based on perspective theory. The main idea is that a point in the object space $O(x_i, y_i, z_i)$ corresponds to a point in the image space $P(u_i, v_i)$ and then converges to the perspective center $C$ behind the image. Hence, the perspective center point $C$, the image point $P$, and the object point $O$ construct a collinear line.

In collinearity equations, nine parameters of the camera should be considered, including the perspective center in the object space coordinates ($C_x$, $C_y$, and $C_z$), the perspective center in the image space coordinates ($C_u$, $C_v$, and $f$), and three rotation angles (azimuth angle θ, roll angle β, and tilt angle τ). Figure 4 depicts the relationship of the three rotation angles between the object and image spaces. It indicates that $X$, $Y$, and $Z$ are the three coordinate axes in the object space, while $U$, $V$, and $F$ are the three coordinate axes in the image space. The azimuth angle θ denotes the angle between the F-axis direction (in the image space) and the $Y$-axis direction (in the object space); the roll angle β expresses the angle between the $V$-axis direction and the $Z$-axis direction; and the tilt angle τ represents the angle between the $F$-axis direction and the $Z$-axis direction. Since the coordinate systems in the object image spaces are different, a transformation/conversion needs to be performed through the rotation angle matrix. The collinearity equations and rotation coefficients can be expressed as

$$U_{dif} - \Delta u = -f \frac{M_1 X_{dif} + M_2 Y_{dif} + M_3 Z_{dif}}{M_7 X_{dif} + M_8 Y_{dif} + M_9 Z_{dif}} \tag{1a}$$

$$V_{dif} - \Delta v = -f \frac{M_4 X_{dif} + M_5 Y_{dif} + M_6 Z_{dif}}{M_7 X_{dif} + M_8 Y_{dif} + M_9 Z_{dif}} \tag{1b}$$

where $\Delta u$ and $\Delta v$ denote errors caused by lens distortion, which can be divided into radial and tangential components. The equations can be written as

$$\Delta u = (u_i - C_u)\left(k_1 r_i^2 + k_2 r_i^4\right) + p_1\left(r_i^2 + 2(u_i - C_u)^2\right) + 2p_2(u_i - C_u)(v_i - C_v) \tag{2a}$$

$$\Delta v = (v_i - C_v)\left(k_1 r_i^2 + k_2 r_i^4\right) + p_2\left(r_i^2 + 2(v_i - C_v)^2\right) + 2p_1(u_i - C_u)(v_i - C_v) \tag{2b}$$

where $r_i$ denotes the distance from any image point to the center of the image; $k_1$ and $k_2$ are the coefficients of radial distortion; $p_1$ and $p_2$ express the coefficients of tangential distortion; $U_{dif}/V_{dif}$ represents the difference between the perspective center $C_u/C_v$ and any image point $u_i/v_i$ projecting in the $u/v$ coordinate system (the image space); f denotes the distance from the perspective center to the image or the equivalent focal length; $X_{dif}/Y_{dif}/Z_{dif}$ expresses the difference between the perspective center $C_x/C_y/C_z$ and any object point $x_i/y_i/z_i$ projecting in the $x/y/z$ coordinate system (the object space). The parameters of the rotation angle matrix ($M_1 \sim M_9$) are composed of the azimuth angle $\theta$, the roll angle $\beta$, and the tilt angle $\tau$ [47], i.e.,

$M_1 = -\cos\tau \cos\theta - \sin\tau \cos\beta \sin\theta$;
$M_2 = \cos\tau \sin\theta - \sin\tau \cos\beta \cos\theta$;
$M_3 = -\sin\tau \sin\beta$;
$M_4 = \sin\tau \cos\theta - \cos\tau \cos\beta \sin\theta$;
$M_5 = -\sin\tau \sin\theta - \cos\tau \cos\beta \cos\theta$;
$M_6 = -\cos\tau \sin\beta$;
$M_7 = -\sin\beta \sin\theta$;
$M_8 = -\sin\beta \cos\theta$;
$M_9 = \cos\beta$.

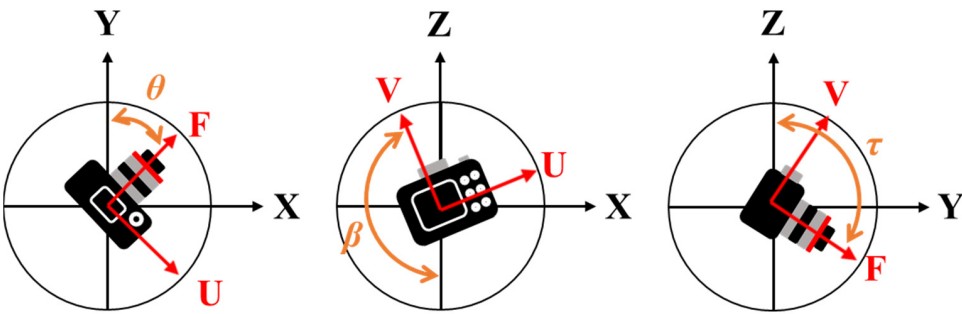

**Figure 4.** Schematic diagram of the rotation angles (i.e., $\theta$, $\beta$, and $\tau$ for azimuth, roll, and tilt, respectively) among the collinear parameters.

### 3.1.2. Image Matching

Image matching is an important part of LSPIV. The basic concept is to search similar features on the surface of a river, e.g., a ripple in the first image at time t and that with a movement in the second image at time $t + \Delta t$. Therefore, an interrogation area (IA) and a search area (SA) in the first and second images should be determined. If the flow direction is uncertain, the SA can be centered on the IA of the first image and then expanded. When the flow condition is constant and known, the SA can be extended along its direction, reducing the ranges of the SA with improved computational efficiency (Figure 5). Later, starting with time $t + \Delta t$, an IA can be established in the SA of the second image by a progressive movement and comparison to the old one. The position with the highest similarity can be regarded as the new position of the IA after $\Delta t$.

Several criteria including the correlation coefficient, covariance, and least squares have been used for image matching. In this study, the correlation coefficient was adopted [33,46].

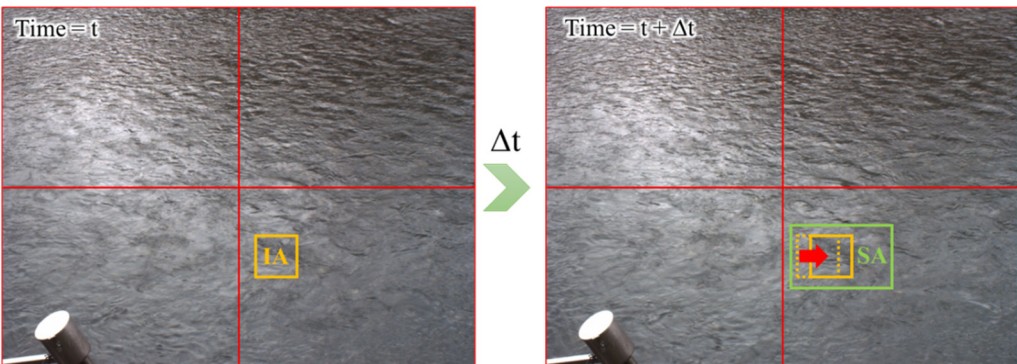

**Figure 5.** Schematic sketch of image matching.

### 3.1.3. Surface Velocity and River Discharge

The surface velocity and river discharge in a region of interest (ROI) can be estimated after the aforementioned steps. In general, a visible range should be determined while setting up the camera. The camera first detects the GCPs through a horizontal rotation so that the nine parameters in the collinearity equations can be resolved. The camera is then rotated back to the ROI for observations. Notice that the azimuth angle ($\theta$) does not affect the analysis of surface velocity, as demonstrated in previous studies [16,48]. Once the parameters are obtained, image acquisition and river surface velocity measurements using LSPIV can be initiated. The image matching step is utilized to find the moving ripple on the river surface. The coordinates in the image space are mapped to the object space through the collinearity equations. Note that only the X and Y coordinates require the transformation (since the water level can be obtained from the gauge station). A simple computation of the moving distance over the traveling time returns the river surface velocity.

In this study, three groups of images were recorded each time, and five consecutive pictures were taken in each group. In other words, the surface velocities were calculated three times and averaged. Note that image matching was performed directly to analyze the surface velocity after acquiring the images. In this step, the IA was 50 pixels × 50 pixels, and the SA was 50% of the IA (i.e., 25 pixels × 25 pixels). The oblique images were not corrected to orthophotos, which provides the advantage of requiring less computation time for LSPIV. Meanwhile, a large number of interpolations can be avoided, reducing noise and image matching errors [45]. Indeed, the oblique image would generate more velocity vectors in the near field and fewer velocity vectors in the far field. The surface velocity field in the near-field image was composed of up to 660 vectors, while the velocity distribution in the far-field image was composed of up to 330 vectors. When calculating the mean surface velocity, a distance-based weighting procedure was used to obtain the correct velocity.

Furthermore, the cross-sectional mean velocity was estimated using the so-called index velocity method. The index velocity method relates the mean velocity to the surface velocity by a site-specific constant k. In this study, the average surface velocity was measured via LSPIV, and the cross-sectional mean velocity was obtained by the flow meter using the two-point method. Typically, the value of k is about 0.85 for a normal flow condition and 0.93 during flood events [18]. Later, the cross-sectional area can be calculated using the water level and bathymetric data. The river discharge as a product of the mean velocity and the cross-sectional area can finally be obtained.

A flow chart for the overall streamflow measurement (surface velocity and river discharge) with LSPIV is shown in Figure 6. The in-house developed LSPIV program was written with MATLAB (2019b, 64-bit) and executed on a Windows 7 operating system. The computer was equipped with an Intel Core i7-4770 3.4 GHz processor with 32 GB DDR3 memory. The analysis of surface velocity within a time interval takes about 10 s to complete the computation.

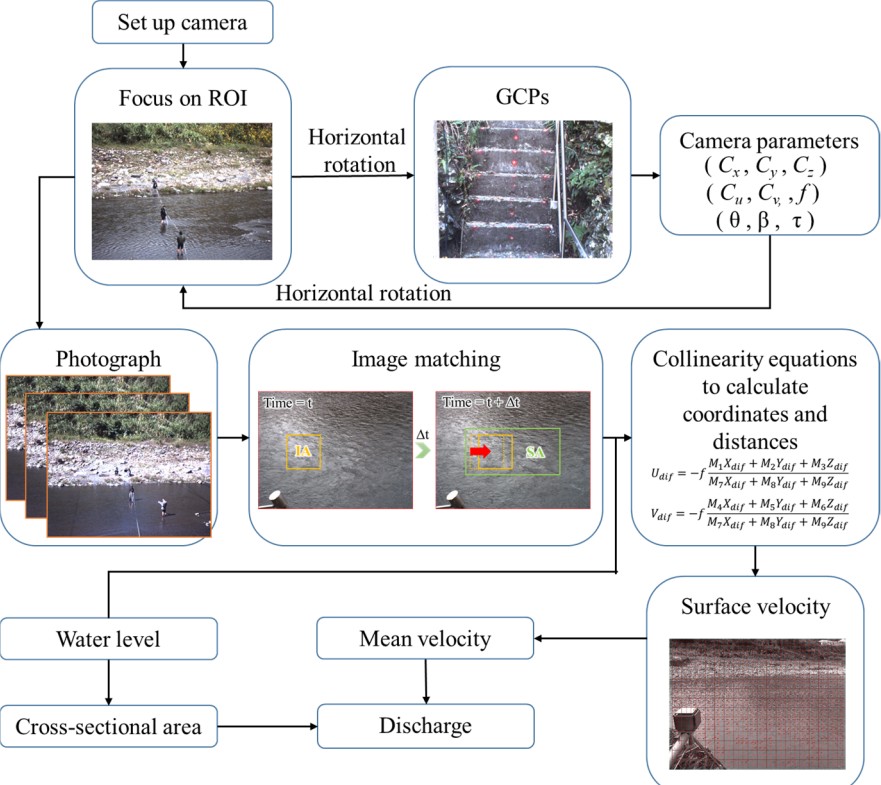

**Figure 6.** Flow chart for the measurement of surface velocity using LSPIV and the estimation of river discharge.

*3.2. Uncertainty Assessment: Monte Carlo Simulations*

The Monte Carlo simulations (MCS) has been widely used for uncertainty assessment in many different fields, e.g., hydrology and measurement studies [42,49–53]. Based on the theory of probability statistics, this method solves and evaluates the uncertainty of a mathematical problem with a large number of randomly generated samples.

The Monte Carlo simulations used in this study consisted of three major steps. The first step was to determine the influential factors (i.e., the GCPs and associated camera parameters through the solution of the collinearity equations); the second step was to obtain a probability distribution function for the random variables; and the third step was to randomly generate a large number of samples using the probability density function.

In this study, uncertainty in the LSPIV measurement (surface velocity and river discharge) due to the GCPs (and nine associated camera parameters) was analyzed. A total of 29 GCPs were placed on the ladder of the gauging platform. For the coordinates of the GCPs, three repeated measurements were carried out, which returned a small standard error (SE) of less than 10 mm. Note that continuous efforts have been made to improve the accuracy of experimental instruments over the decades. Here, we considered 3 times and 0.3 times the standard error in the GCP measurement to represent worse and better quality (due to the old- and new-generation instruments or other factors), respectively. To quantify the influence of GCPs, a large number of samples were randomly generated by a superposition of the original coordinates and the Gaussian-distributed standard errors (i.e., 30 mm, 10 mm, and 3 mm). The camera parameters were then solved using the collinearity equations for each GCP sample. To reduce the large amount of computation, frequency analysis was also performed for these camera parameters (including the beta, gamma, normal, Weibull, and log Pearson type III probability functions determined by the standard error and correlation coefficient). In other words, these nine camera parameters with their optimal probability functions representing GCPs with various standard errors could be directly used for the uncertainty assessment (see Figure 7).

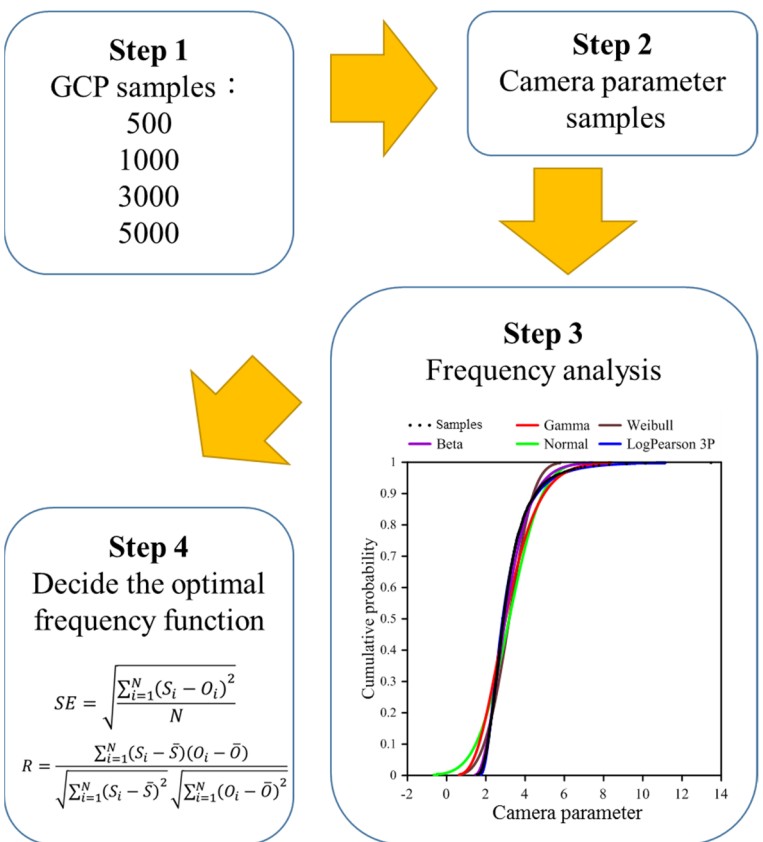

**Figure 7.** The procedure for determining the optimal probability distribution function in the frequency analysis.

## 4. Results and Discussion

### 4.1. Streamflow Measurement Using LSPIV

There were four field experiments conducted on 3 May, 26 July, 1 November, and 3 December in 2020. The measured surface velocities were compared with those obtained from the flow meter for validation.

To begin the LSPIV measurement, the GCPs were first placed on the ladder of the gauging platform, as illustrated in Figure 8, where 11 and 18 GCPs were deployed for the near-field and far-field cameras (Figure 8a,b), respectively. Each GCP was measured three times using a total station (Nikon NE101), returning a standard error of about 10 mm. The cameras were rotated to take the images of the GCPs to calculate the nine camera parameters based on the collinearity equations. The dual cameras were rotated backwards horizontally to observe the ROI (see Figure 8c,d).

The parameters of the near-field and far-field cameras in the four field experiments are shown in Table 1. The camera parameters for each experiment were slightly different because the cameras were installed on the experimental day. In the image space, the coordinates of the perspective center were $C_u$ = 812 pixels and $C_v$ = 617 pixels. The focal length f was 0.018 m for the far-field camera, and it ranged from 0.012 m to 0.017 m for the near-field camera. In the object space, the coordinates $C_x$, $C_y$, and $C_z$ ranged from −0.393 m to 0.373 m, −1.749 m to 0.41 m, and 0.778 m to 1.744 m, respectively. Both the azimuth angle θ and roll angle β were close to 180° (from 168.15° to 189.17°). The tilt angle τ of the near-field camera (between 110.72° and 115.46°) was larger than that of the far-field camera (96.63° to 99.28°), giving an ROI close to the right bank for the near-field camera (Figure 8c) and an ROI near the left bank for the far-field camera (Figure 8d).

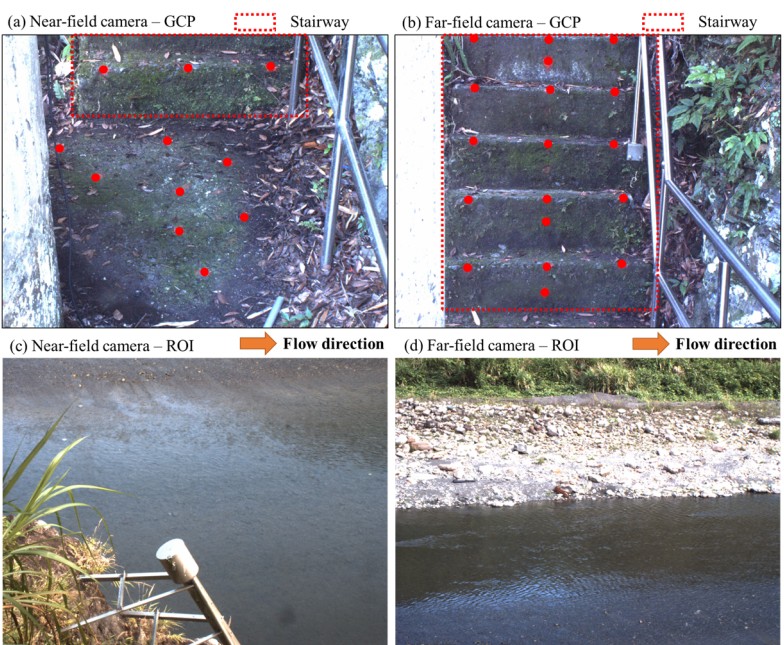

**Figure 8.** (**a**,**b**) GCPs and (**c**,**d**) ROIs from the view of the near-field and far-field cameras.

**Table 1.** Parameters of the near-field (NF) and far-field (FF) cameras in the four field experiments.

| Date/Camera | | $C_u$ (Pixels) | $C_v$ (Pixels) | $f$ (m) | $C_x$ (m) | $C_y$ (m) | $C_z$ (m) | $\theta$ (Degrees) | $\beta$ (Degrees) | $\tau$ (Degrees) |
|---|---|---|---|---|---|---|---|---|---|---|
| 3 May 2020 | FF | 812 | 617 | 0.018 | −0.16 | −1.595 | 1.648 | 178.83 | 184.16 | 99.28 |
| | NF | 812 | 617 | 0.015 | −0.261 | −0.629 | 1.064 | 168.15 | 186.42 | 111.77 |
| 26 July 2020 | FF | 812 | 617 | 0.018 | −0.189 | −1.696 | 1.605 | 178.65 | 184.09 | 97.38 |
| | NF | 812 | 617 | 0.014 | 0.070 | 0.410 | 0.778 | 175.92 | 178.02 | 110.72 |
| 1 November 2020 | FF | 812 | 617 | 0.018 | −0.393 | −1.675 | 1.648 | 178.26 | 189.17 | 96.63 |
| | NF | 812 | 617 | 0.017 | −0.145 | −0.634 | 1.133 | 160.76 | 184.64 | 115.46 |
| 3 December 2020 | FF | 812 | 617 | 0.018 | 0.373 | −1.749 | 1.744 | 178.86 | 178.25 | 98.76 |
| | NF | 812 | 617 | 0.012 | 0.017 | −0.080 | 1.132 | 176.70 | 184.84 | 112.26 |

The surface velocity field and contour in the ROI measured by LSPIV are shown in Figure 9. Note that the origin of the coordinates in Figure 9 is the control point on the left-bank embankment, which is opposite to the water gauge station. The coordinates of the camera are (0, 70). As can be seen, the main flow direction was from left to right. Some variations may occur in the lower or middle part (Y = 5 m to 15 m) of Yufeng Creek, e.g., the flow patterns measured on 26 July and 1 November (see Figure 9b,c). Moreover, the maximum velocity was about 1.7 m/s near the shore area (Figure 9b). Furthermore, contour plots for the magnitude of surface velocities are shown in Figure 9e–h. Interestingly, the flow conditions of Yufeng Creek could be divided into three regions (i.e., Y = 5 m to 10 m, 10 m to 25 m, and 25 m to 35 m), with higher velocities near the shore and lower velocities in the middle of the river. The reason is that a shoal that formed around April significantly affected the flow pattern. Note that these discontinuous changes in the velocity field (see Figure 9e,f) from LSPIV are quite common. Typically, the river flows affected by upstream discharges and the bottom topography are in a highly turbulent condition. The LSPIV measurement estimated the velocity through the analysis of instantaneous ripples (or objects) on the surface, possibly capturing the occurrence of discontinuous variations in the velocity field (see contour maps in Figure 9). Similar results were also reported in previous works [25,30].

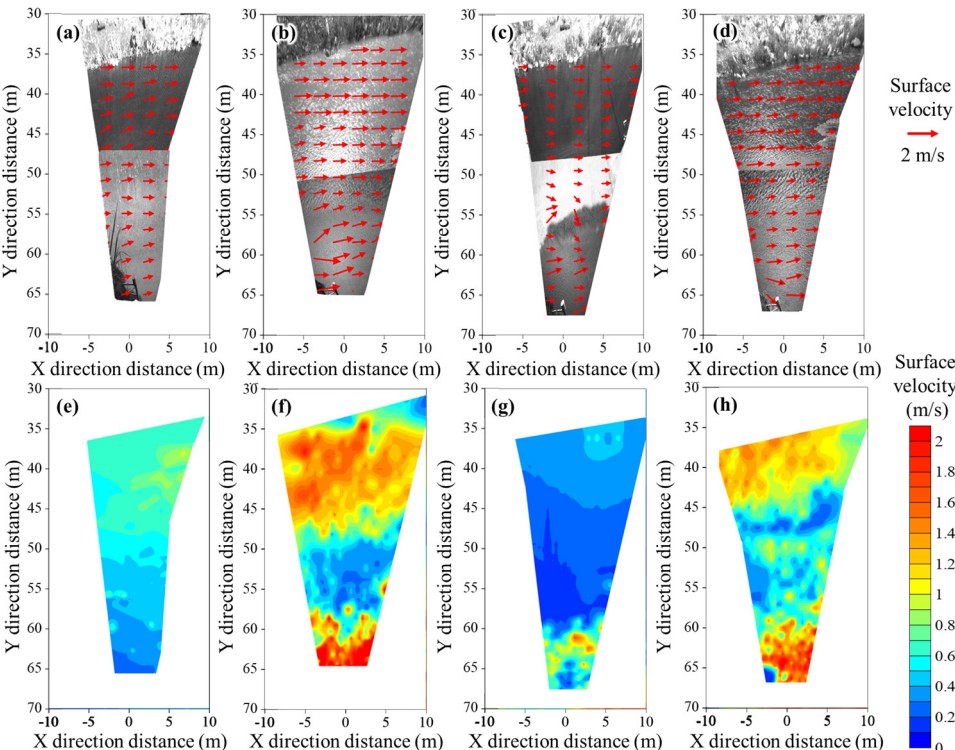

**Figure 9.** Surface velocity fields and contours measured using LSPIV at (**a**,**e**) 9:00 on 3 May 2020, (**b**,**f**) 16:00 on 26 July 2020, (**c**,**g**) 11:00 on 1 November 2020, and (**d**,**h**) 14:00 on 3 December 2020.

The surface velocities along the cross-section obtained from the LSPIV and flow meter measurements are compared in Figure 10. The spatial distributions of the river flows were in good agreement, indicating lower velocities in the middle part of the river. As mentioned above, this was attributed to local topography effects (i.e., higher riverbed or shallower depth due to the shoal). Furthermore, the time series of averaged surface velocities measured by LSPIV and the flow meter during the four field experiments are shown in Figure 11. While some obvious oscillations were found in the continuous LSPIV measurement, the averaged surface velocities obtained from both methods were in reasonable agreement. In the 10 h experiment on 3 May, for example, the temporal mean of the averaged surface velocity was about 0.5 m/s (0.528 m/s from the flow meter and 0.485 m/s from LSPIV). Moreover, a scatter plot that demonstrates the correlation between pairs of the measured results is shown in Figure 12. The averaged surface velocities measured by LSPIV and the flow meter were highly correlated, giving the regression Y = 0.916X + 0.015, where X and Y represent the results from the flow meter and LSPIV measurements, respectively. The regression with $R^2$ = 0.55 and a *p*-value ~$10^{-8}$ is statistically significant under a 95% confidence interval. Furthermore, in terms of accuracy, the mean absolute error (MAE) and root mean square error (RMSE) of the averaged surface velocity are summarized in Table 2. The values of MAE and RMSE ranged from 0.097 m/s to 0.154 m/s and from 0.107 m/s to 0.191 m/s, respectively. Overall, the LSPIV method demonstrated strong reliability in measuring the river surface velocity. Among the four experiments, the best performance was obtained on 3 May 2020. For the remaining three dates, the performances with slight differences were also satisfactory.

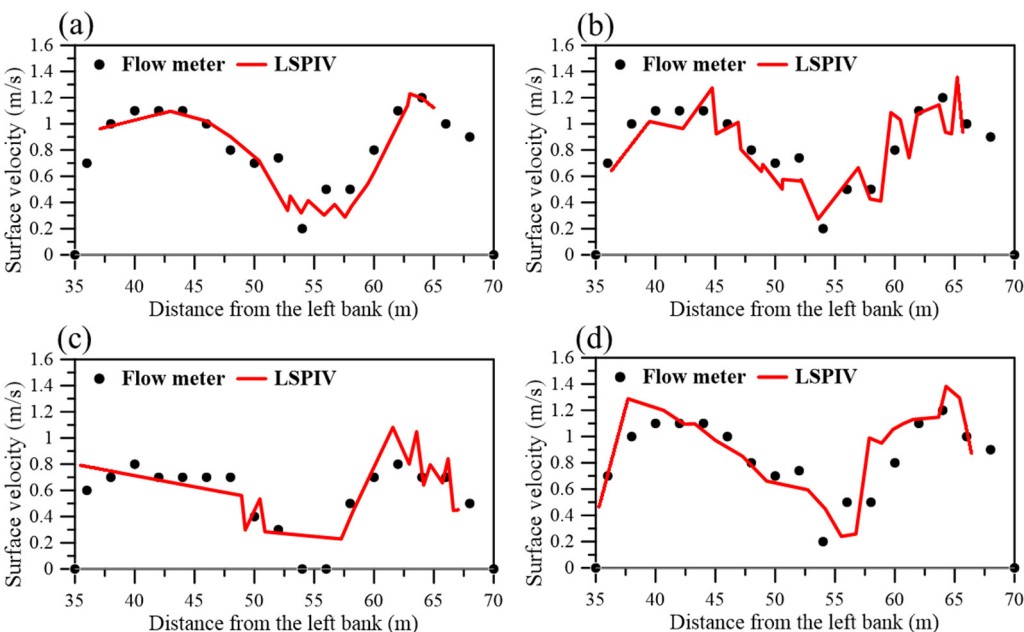

**Figure 10.** Comparison of surface velocities measured by LSPIV and the flow meter on (**a**) 3 May 2020, 10 a.m., (**b**) 26 July 2020, 12 p.m., (**c**) 1 November 2020, 11 a.m., and (**d**) 3 December 2020, 4 p.m.

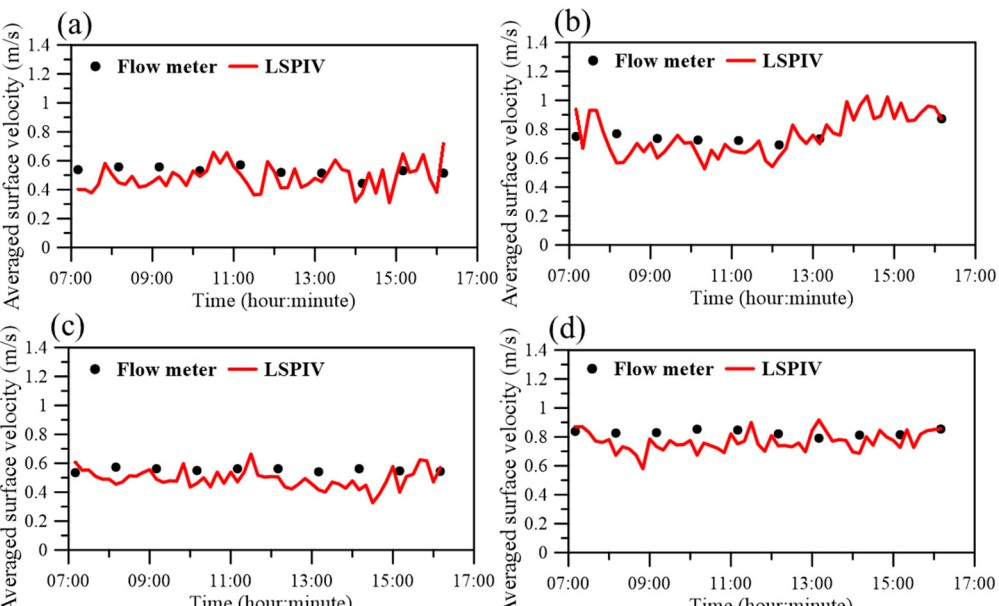

**Figure 11.** Comparison of surface velocities measured by LSPIV and the flow meter on (**a**) 3 May 2020, (**b**) 26 July 2020, (**c**) 1 November 2020, and (**d**) 3 December 2020.

**Table 2.** The averaged surface velocities and two statistical indices (MAE and RMSE) between the results measured by LSPIV ($V_{LSPIV}$) and the flow meter ($V_{FM}$).

| Date | 3 May | 26 July | 1 November | 3 December |
|---|---|---|---|---|
| Max water depth (m) | 0.66 | 0.79 | 0.62 | 0.85 |
| VFM (m/s) | 0.528 | 0.750 | 0.554 | 0.829 |
| VLSPIV (m/s) | 0.485 | 0.665 | 0.492 | 0.758 |
| MAE (m/s) | 0.097 | 0.154 | 0.104 | 0.098 |
| RMSE (m/s) | 0.107 | 0.191 | 0.111 | 0.110 |

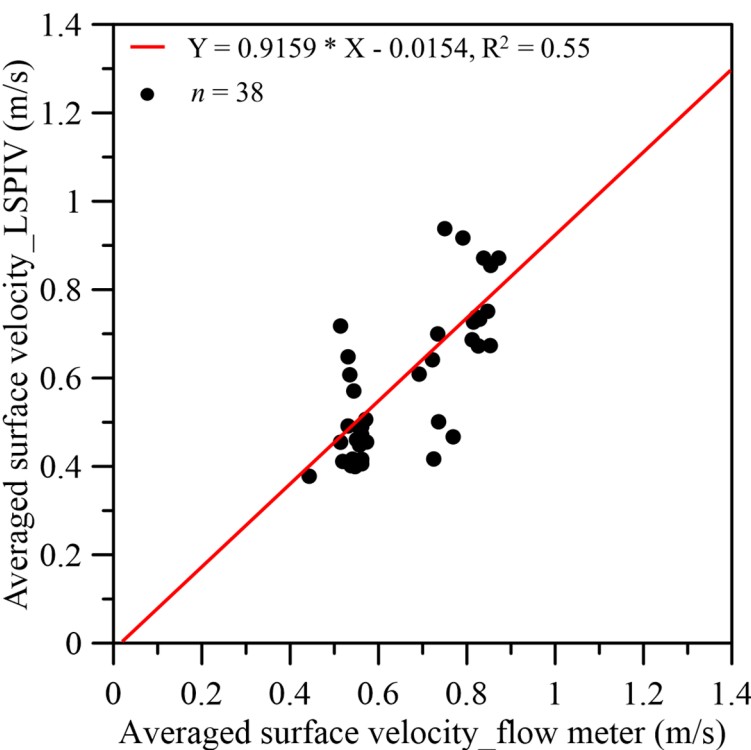

**Figure 12.** Relationship between averaged surface velocities measured by LSPIV and the flow meter.

*4.2. Uncertainty in GCPs and Camera Parameters*

In this study, the field experiment conducted on 3 May 2020 was chosen as the basis for further uncertainty analysis, as the best performance of the LSPIV measurement was obtained on this date. Hence, the original coordinates of GCPs were superimposed with the Gaussian-distributed standard errors (e.g., 10 mm), returning a total of 5000 samples and the corresponding camera parameters through the collinearity equations.

Figures 13 and 14 show the random samples and the fitted cumulative probability distributions of the nine parameters of the near-field and far-field cameras, respectively. Unlike other camera parameters, the coordinates of the perspective center $C_u$ and $C_v$ are integers (i.e., the pixels in the image space), forming several groups in a vertical line for the 5000 GCP samples. The probability distributions considered in this study included the Gumbel, Weibull, beta, normal, and log Pearson type III functions. Two statistical indices, i.e., the standard error SE and correlation coefficient R, were used to determine the optimal probability function for each parameter. All the probability density functions have good correlations, with R > 0.95. The values of the standard errors are listed in Table A1 in the Appendix A. According to their fitness, the parameters and their distribution functions are summarized as follows:

1.　Near-field camera: $C_u$ (normal), $C_v$ (normal), $f$ (log Pearson type III), $C_x$ (beta), $C_y$ (beta), $C_z$ (log Pearson type III), θ (normal), β (normal), and τ (log Pearson type III);
2.　Far-field camera: $C_u$ (log Pearson type III), $C_v$ (normal), $f$ (log Pearson type III), $C_x$ (log Pearson type III), $C_y$ (Weibull), $C_z$ (log Pearson type III), θ (normal), β (normal), and τ (log Pearson type III).

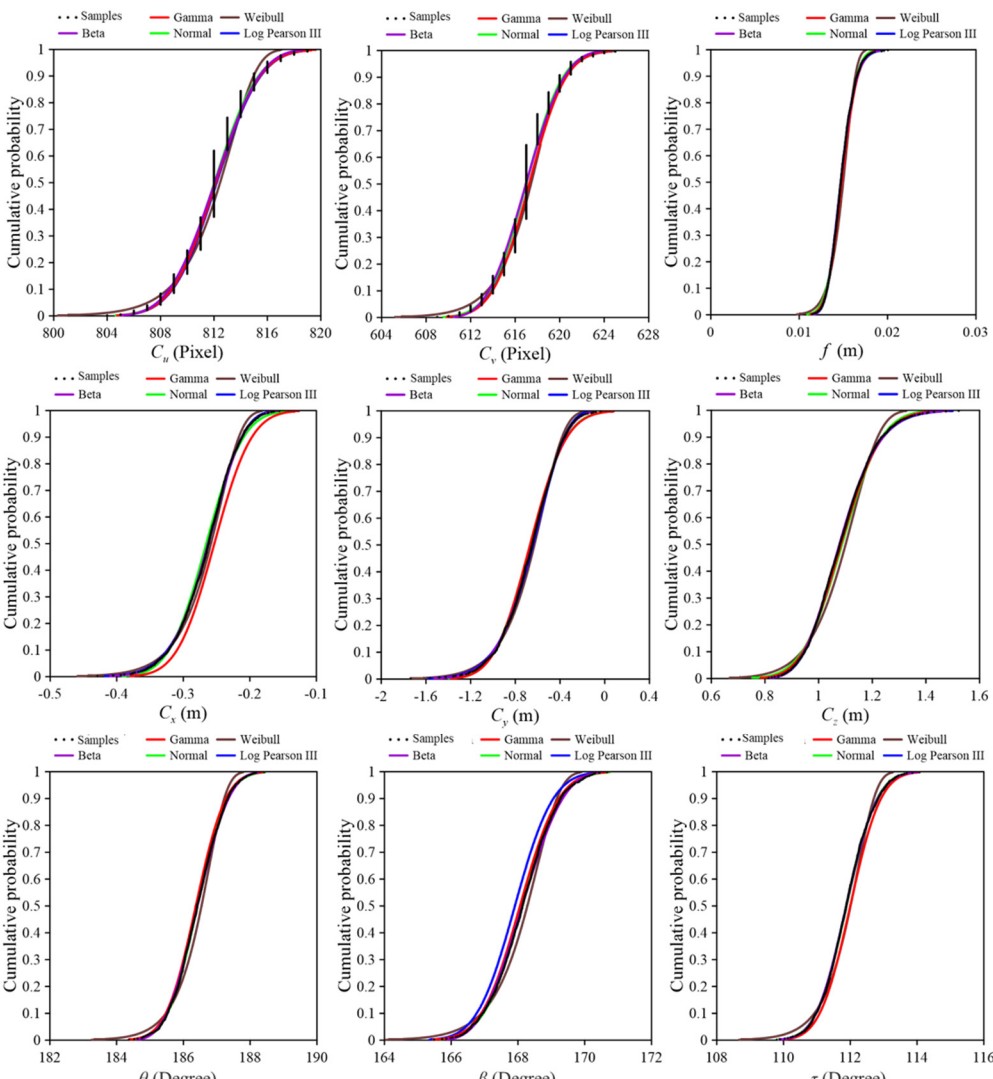

**Figure 13.** Comparison of the samples and various probability distribution functions for the nine parameters of the near-field camera.

### 4.3. Uncertainty in Streamflow Measurement: GCP Measurement Times

A repeated operation of the total station measurement for the coordinates of the GCPs might reduce the uncertainty in LSPIV. In practice, the measurements of GCPs are repeated at least three times and averaged to provide more accurate and less uncertain results for the coordinates of the GCPs. Thus, reliable and successful LSPIV applications for streamflow measurements can be ensured. To analyze the impacts from different operations (with a given measurement accuracy), Monte Carlo simulations and an uncertainty assessment were carried out. In this paper, the GCP measurements subject to a standard error of 10 mm were repeated one, three, five, seven, and nine times. Subsequently, a total of 100 realizations were carried out for the uncertainty assessment, providing a cumulative distribution (e.g., 2.5%, 25%, 50%, 75%, and 97.5%) for these measured surface velocities.

Based on the 100 MCS realizations, Figure 15 presents the uncertainty assessment of the surface velocities from the LSPIV measurement. Note that the light and dark blue areas denote the 95% (2.5% to 97.5%) and 50% (25% to 75%) confidence intervals for the measured velocities. In the meantime, the surface velocity obtained by the flow meter (the black dots) is included for comparison. For the three repeated measurements of GCPs, as an example, the median (or mode) of the surface velocity in these 100 realizations (i.e., LSPIV measurements) was close to 0.485 m/s, again in excellent agreement with the averaged results (0.528 m/s) from the flow meter. In terms of uncertainty, given the

cumulative probabilities of 2.5%, 25%, 75%, and 97.5%, the surface velocities from the LSPIV measurement were 0.44 m/s, 0.46 m/s, 0.50 m/s, and 0.52 m/s, respectively. Due to the uncertainty in GCPs, deviations of −9.3% and 7.2% (lower and upper bounds) from the median (or mode) were obtained for the surface velocities. Figure 16 further compares the surface velocities obtained from LSPIV under different repeated GCP measurements (1, 3, 5, 7, and 9 times with 100 realizations). As shown in Figures 15a and 16, given the cumulative probability of 97.5% (or 99%), a surface velocity of up to 0.612 m/s (or 0.708 m/s) can be found in the case of a single GCP measurement. Clearly, the method of operation has a great influence on uncertainty. The impacts (i.e., a shifted and wider confidence interval) can be alleviated when the measurements of GCPs are repeated. Additionally, the uncertainty in the measured surface velocities will converge to a small range (0.06 m/s) when the GCP measurements are repeated more than three times.

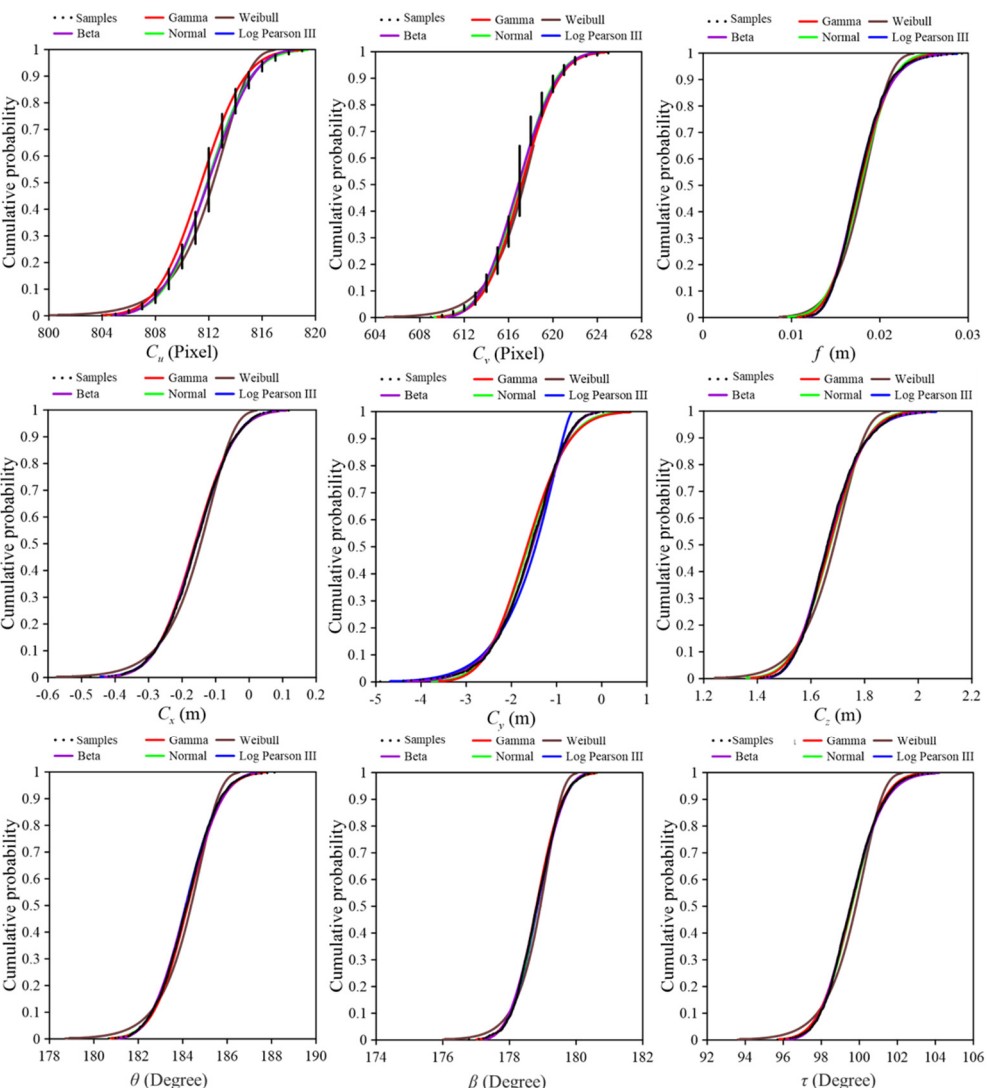

**Figure 14.** Comparison of the samples and various probability distribution functions for the nine parameters of the far-field camera.

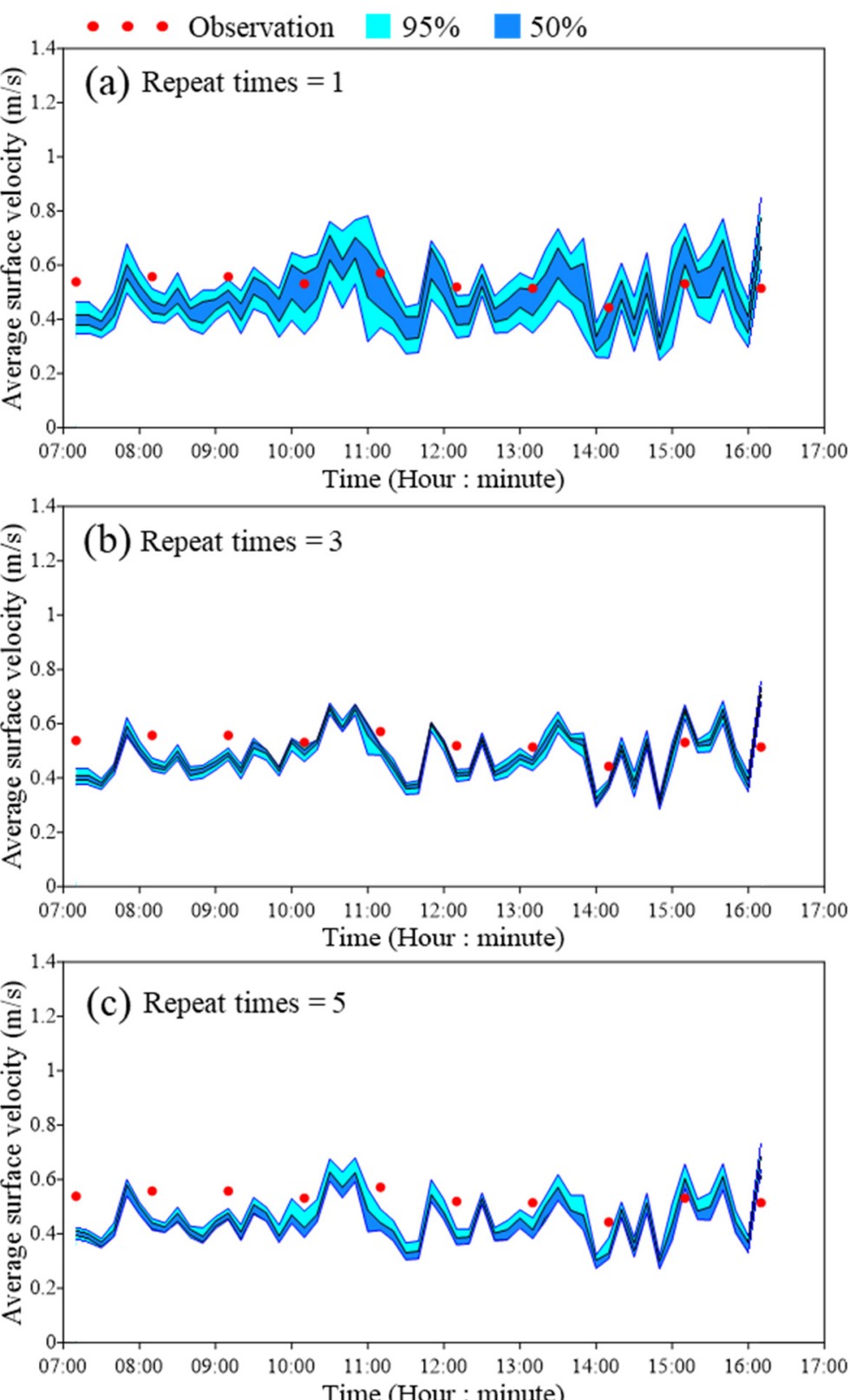

**Figure 15.** Uncertainty analysis of the averaged surface velocities from LSPIV based on Monte Carlo simulations: (**a**) one, (**b**) three, and (**c**) five GCP measurements. Note that the red dots represent the average surface velocity measured by the flow meter; the light blue area denotes the 95% (i.e., from 2.5% to 97.5%) confidence interval; and the dark blue area expresses the 50% (i.e., from 25% to 75%) confidence interval.

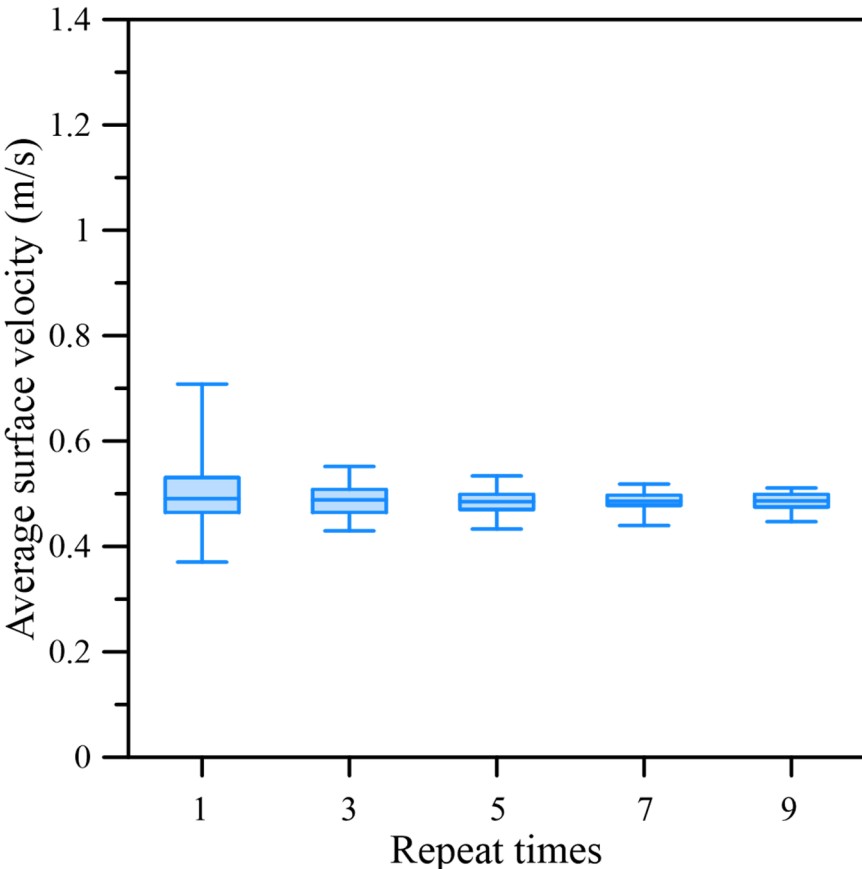

**Figure 16.** Box plot for the averaged surface velocities from LSPIV with different GCP measurement times.

*4.4. Uncertainty in Streamflow Measurement: GCP Measurement Accuracy*

The uncertainty in LSPIV streamflow measurements due to the measurement accuracy of GCPs was explored. A range of standard errors in the GCP measurements, i.e., 30 mm, 10 mm, and 3 mm, were considered in this study. To avoid interference from the operations, the GCP measurements were repeated three times.

Figure 17a–c shows the surface velocities obtained from LSPIV measurements for different GCP accuracies, i.e., SE = 30 mm, 10 mm, and 3 mm. The medians of the surface velocities for these three scenarios were 0.488 m/s, 0.485 m/s, and 0.483 m/s, respectively. For the SE of 30 mm, the lower (2.5%) and upper (97.5%) bounds of the surface velocities were 0.425 m/s and 0.560 m/s, giving a wider range (0.135 m/s) for the 95% confidence interval. The measured surface velocities would be reduced to about 0.05 m/s if the SE could be improved to 10 mm or 3 mm. Based on a large number of random samples from the Monte Carlo simulations, a possible impact from larger but less frequent errors (i.e., an extremely low probability for significant under- or over-estimations of the surface velocities) was revealed. A greater uncertainty (especially in the upper bounds) attributed to the larger standard error was found, although the median of the surface velocities from LSPIV was still quite close to that measured by the flow meter. Overall, the measurement accuracy of GCPs plays an important role in LSPIV surface velocity measurement.

Furthermore, the surface velocity was converted to the cross-sectional mean velocity with k = 0.87 based on the measured data from four field experiments. Legleiter et al. [15] measured five rivers and obtained the constant k in a range from 0.819 to 0.927, supporting the reasonable setting of k in this study. The river discharge was then obtained by multiplying the depth-averaged velocity and the cross-sectional area. Figure 17d–f present the histograms for river discharges.

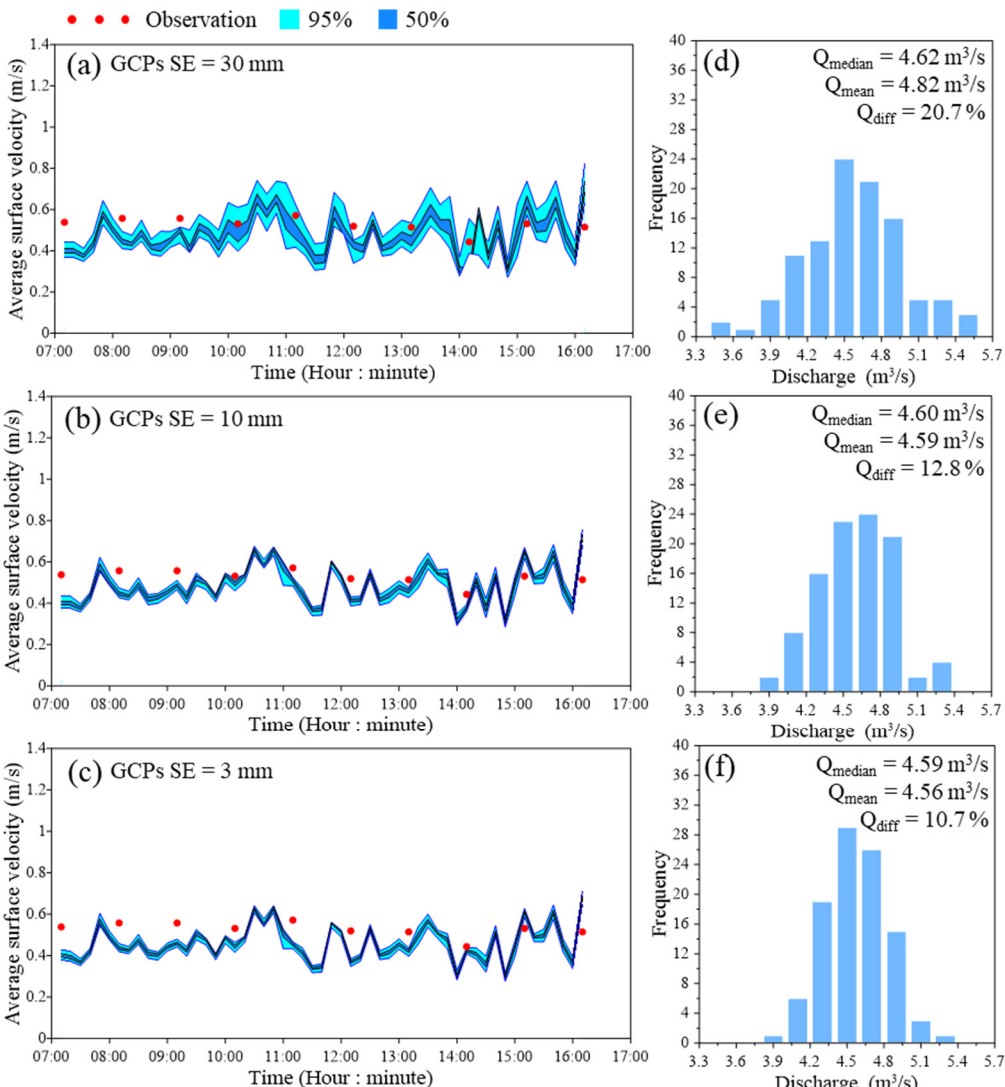

**Figure 17.** Uncertainty analysis of the averaged surface velocities from LSPIV and the estimated river discharges based on Monte Carlo simulations: (**a**,**d**) SE = 30 mm, (**b**,**e**) SE = 10 mm, and (**c**,**f**) SE = 3 mm in GCPs. Note that the red dots represent the average surface velocity measured by the flow meter; the light blue area denotes the 95% (i.e., from 2.5% to 97.5%) confidence interval; and the dark blue area expresses the 50% (i.e., from 25% to 75%) confidence interval.

Based on the MCS, the measured discharges ranged from 3.3 m³/s to 5.7 m³/s for the case of SE = 30 mm. While the mean discharge $Q_{mean}$ was 4.82 m³/s, the mode of discharges with an occurrence probability of 24% was about 4.62 m³/s (Figure 17d). Note that the mean of the surface velocities was potentially influenced by extreme values. The median (or mode) would be a more appropriate way to represent the surface velocities for comparison. The uncertainty in the discharge caused by the GCP measurement accuracy can be expressed as half of the normalized confidence interval, i.e., $Q_{diff} = (Q_{97.5\%} - Q_{2.5\%})/Q_{mean}/2 = 20.7\%$, where $Q_{97.5\%}$ and $Q_{2.5\%}$ denote discharges with cumulative probabilities of 97.5% and 2.5%, respectively. In the case of SE = 3 mm, the measured discharges were distributed mainly in a range between 4.5 m³/s and 4.7 m³/s (with occurrence probabilities of 29% and 26% or a total of 55% in the distribution), returning a mean discharge of 4.59 m³/s and a normalized half confidence interval of 10.7% (Figure 17f). As the accuracy of the GCP measurements increased (SE = 30 mm, 10 mm, and 3 mm), the uncertainty in the LSPIV streamflow measurements ($Q_{diff}$ =20.7%, 12.8%, and 10.7%) decreased, returning

median discharges ($Q_{median}$ = 4.62 m$^3$/s, 4.6 m$^3$/s, and 4.59 m$^3$/s) closer to those (4.59 m$^3$/s) obtained from the flow meter.

The results of the present study (cases of SE = 10 mm and SE = 3 mm) are similar to the findings of Le Coz et al. [42], where the errors in the camera parameters were smoothed out by employing more GCPs. The $Q_{diff}$ was about 12% when the number of GCPs reached 19. Additionally, Schweitzer and Cowen [54] demonstrated that an accurate GCP-based georeferencing method was able to reduce uncertainty in the streamflow measurement by a factor of five or more in comparison to the direct method. Overall, both previous and current works clearly imply that a high-precision instrument for GCP measurement is necessary.

*4.5. Limitations and Future Work*

In this study, for the GCPs, the dimensions of the layout were about 3 m (width) × 7 m (length) × 5 m (height). Based on three repeated measurements, the standard error of 10 mm indicated a measurement accuracy of about 1/300 to 1/1000. For the ROI, the image resolution was 1624 × 1234 pixels for a measurement distance over 5 m or 40 m. The uncertainty in LSPIV streamflow measurements was about 12%. Overall, the results imply that a person who completes the GCP measurements using a precise total station after simple training would be able to obtain accurate streamflow measurements with a similar degree of uncertainty to this study. If the ROI area and the image resolution are different, the uncertainty in the LSPIV streamflow (surface velocity and river discharge) measurements would be changed. It is recommended that a standard measurement procedure be applied to check if there are any errors from other sources after the three repeated measurements and to confirm the GCP measurement accuracy.

In addition to the camera parameters, the uncertainty in the LSPIV streamflow measurements may result from other influential factors including the image resolution, orthorectification, interrogation area (IA), and search area (SA). For example, Fleit and Baranya [55] examined the influence of the SA (i.e., static SA, adaptive SA, isotropic SA, and anisotropic SA) on the LSPIV surface velocity measurement. Rozos et al. [41] also investigated the effects of IA size and the associated uncertainty in the LSPIV surface velocity measurement using Monte Carlo simulations. The impacts of these factors on the LSPIV streamflow measurement of a watershed area will be considered in future work.

**5. Conclusions**

Streamflow measurements, which provide essential data on river discharges, play a critical role in hydro-environmental research. Conventional workflows for effective streamflow measurements can be quite tedious, time-consuming, difficult, and dangerous [6]. Thus, through continuous efforts over the past decade, image-based velocimetry algorithms [18,19,21] have been developed to provide a cost-effective, rapid, and secure monitoring tool for streamflow measurements (including surface velocity and river discharge), i.e., large-scale particle image velocimetry (LSPIV). Nevertheless, the surveys of ground control points (GCPs) that affect the camera parameters through collinearity equations might impose uncertainty on the LSPIV streamflow measurement results [43,44].

The purpose of this study was to explore the uncertainty in image-based streamflow measurements with the main focus on ground control points. The degree of uncertainty in the LSPIV streamflow measurement was quantified using Monte Carlo simulation (MCS), in which a large number of camera parameters obtained from the collinearity equations and ground control points were randomly sampled under different standard errors. The study area was Yufeng Creek, which is upstream of the Shimen Reservoir in Northern Taiwan. A system of dual cameras was set up on the platform of a gauge station to analyze the surface velocity.

To ensure the accuracy of image-based LSPIV, through four field experiments, a comparison with the conventional measurement using a flow meter was also conducted. The results showed that the ranges of the mean absolute error (MAE) and root mean squared

error (RMSE) were 0.097 m/s to 0.154 m/s and 0.107 m/s to 0.191 m/s, respectively. The field experiment conducted on 3 May 2020 was chosen as the basis for further uncertainty analysis since the best performance of the LSPIV measurement was obtained on this date. The original coordinates of the GCPs were superimposed with the Gaussian-distributed standard errors (e.g., 10 mm), returning a total of 5000 samples and the corresponding camera parameters through the solution of the collinearity equations. To reduce the large amount of computation, a common frequency analysis was also performed for these camera parameters (including beta, gamma, normal, Weibull, and log Pearson type III probability functions, determined by the standard errors and correlation coefficients).

Lastly, the uncertainty in the LSPIV measurement influenced by GCPs (due to various measurement times and accuracies) was quantified. For the three repeated measurements of GCPs, as an example, the median (or mode) of the surface velocity (close to 0.485 m/s) in these 100 realizations (i.e., LSPIV measurements) was in excellent agreement with the averaged results (0.528 m/s) from the flow meter. In terms of uncertainty, a range from 0.44 m/s to 0.52 m/s (or $-9.3\%$ to 7.2%) was obtained for the lower (2.5%) and upper (97.5%) bounds of the averaged surface velocities. The method of operation in GCP measurements interferes with uncertainty, e.g., a shifted and wider range of the confidence interval. In terms of different measurement accuracies (i.e., SE = 30 mm, 10 mm, and 3 mm), the medians of the surface velocities for the three scenarios were 0.488 m/s, 0.485 m/s, and 0.483 m/s, respectively. In the case of SE = 30 mm, the lower and upper bounds of the surface velocities were 0.425 m/s and 0.560 m/s, giving a wider range for the 95% confidence interval. Furthermore, the river discharge was obtained using the cross-sectional area and the mean velocity. As the accuracy of the GCP measurements increased (i.e., SE = 30 mm, 10 mm, and 3 mm), overall, the uncertainty in the LSPIV streamflow measurements decreased (i.e., $Q_{diff}$ = 20.7%, 12.8%, and 10.7%), returning median discharges ($Q_{median}$ = 4.62 m$^3$/s, 4.60 m$^3$/s, and 4.59 m$^3$/s) closer to those (4.59 m$^3$/s) obtained from the flow meter.

Overall, the accuracy of GCPs plays a critical role in controlling the uncertainty in LSPIV streamflow measurements, which is consistent with the results of some previous studies [42,54]. The present study further indicates that the uncertainty in LSPIV measurements of the surface velocity of a river can be greatly reduced if the coordinates of the control points are measured and averaged with three repetitions. The accuracy of LSPIV stream velocity measurements is satisfactory if the standard error for the coordinates of the control points is less than 10 mm. LSPIV systems in a fixed study site have shown that they are capable of carrying out long-term continuous monitoring (even under severe weather conditions) [16]. Finally, in addition to the camera parameters, uncertainty in LSPIV measurements may result from other influential factors [41,55]. The impacts of these factors on the LSPIV measurement of a watershed area will be considered and reported in the near future.

**Author Contributions:** Conceptualization, W.-C.L., W.-C.H. and C.-C.Y.; methodology, W.-C.L. and C.-C.Y.; validation, W.-C.H.; formal analysis, W.-C.L., W.-C.H. and C.-C.Y.; investigation, W.-C.L., W.-C.H. and C.-C.Y.; resources, W.-C.L.; writing—original draft preparation, W.-C.L., W.-C.H. and C.-C.Y.; writing—review and editing, W.-C.L., W.-C.H. and C.-C.Y.; visualization, W.-C.H.; supervision, W.-C.L. and C.-C.Y.; project administration, W.-C.L.; funding acquisition, W.-C.L. All authors have read and agreed to the published version of the manuscript.

**Funding:** This study was supported by the Ministry of Science and Technology (MOST), Taiwan, under grant nos. 107-2119-M-239-002 and 108-2119-M-239-001. The financial support is greatly appreciated.

**Institutional Review Board Statement:** Not applicable.

**Informed Consent Statement:** Not applicable.

**Data Availability Statement:** The data and material used in this study can be requested by the readers.

**Conflicts of Interest:** All the authors declare that they have no conflict of interest.

## Appendix A

**Table A1.** Standard errors of various probability distributions for the parameters of the near-field and far-field cameras.

| Parameter | Near-Field Camera | | | | | Far-Field Camera | | | | |
|---|---|---|---|---|---|---|---|---|---|---|
| | Beta | Gamma | Normal | Log Pearson III | Weibull | Beta | Gamma | Normal | Log Pearson III | Weibull |
| $C_u$ (Pixels) | 0.5064 | 0.4716 | 0.4454 | 0.4964 | 0.7246 | 0.4924 | 0.6789 | 0.4223 | 0.4151 | 0.7273 |
| $C_v$ (Pixels) | 0.5715 | 0.5750 | 0.5064 | 0.5201 | 0.6487 | 0.5609 | 0.5482 | 0.4903 | 0.5005 | 0.6654 |
| $f$ (m) | $6 \times 10^{-5}$ | $12 \times 10^{-5}$ | $15 \times 10^{-5}$ | $5 \times 10^{-5}$ | $35 \times 10^{-5}$ | $15 \times 10^{-5}$ | $27 \times 10^{-5}$ | $45 \times 10^{-5}$ | $11 \times 10^{-5}$ | $76 \times 10^{-5}$ |
| $C_x$ (m) | 0.0013 | 0.0129 | 0.0042 | 0.0014 | 0.0059 | 0.0035 | 0.0042 | 0.0035 | 0.0022 | 0.018 |
| $C_y$ (m) | 0.0134 | 0.0334 | 0.0306 | 0.0151 | 0.0269 | 0.0475 | 0.1521 | 0.1255 | 0.1131 | 0.0473 |
| $C_z$ (m) | 0.0030 | 0.0072 | 0.0123 | 0.0027 | 0.0295 | 0.0034 | 0.0100 | 0.0130 | 0.0032 | 0.0320 |
| $\theta$ (Degrees) | 0.0289 | 0.0533 | 0.0167 | 0.0209 | 0.1611 | 0.0731 | 0.0659 | 0.0400 | 0.0460 | 0.2977 |
| $\beta$ (Degrees) | 0.0460 | 0.0718 | 0.0195 | 0.2213 | 0.2115 | 0.0392 | 0.0266 | 0.0218 | 0.0256 | 0.1393 |
| $\tau$ (Degrees) | 0.0178 | 0.1120 | 0.0189 | 0.0176 | 0.1726 | 0.0670 | 0.0680 | 0.0714 | 0.0341 | 0.3639 |

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
