# Peer review of "Uncertainty Analysis for Image-Based Streamflow Measurement: The Influence of Ground Control Points"

_water, doi:10.3390/w15010123_

Round 1
Reviewer 1 Report
The manuscript is well written, and the results are promising. Following are my queries/suggestions:
1) In case of image matching, if the ripple sizes are modified, how it is identified. Because generally the ripples are dynamic in nature. This needs to be elaborated.
2) In Figure 11, there is no meaning of equation Y=X (R=0.74). This has to be removed. Further, what is confidence limit for the fitting?
3) Line 401, how many times the measurements on GCPs are taken. Please indicate instead of several times.
4) What is the reason for median of the surface velocities are considered in the comparison and not the mean?
5) What would be the optimum number of GCP's after which the error becomes negligible?
Author Response
Response to Reviewer 1:
Comments and Suggestions for Authors
The manuscript is well written, and the results are promising. Following are my queries/suggestions:
Response:
We thank the reviewer 1 for the supports of this study. Especially, we appreciate these valuable suggestions for further improvement of the quality of this study.
1) In case of image matching, if the ripple sizes are modified, how it is identified. Because generally the ripples are dynamic in nature. This needs to be elaborated.
Response 1:
The reviewer’s comment is a good point. Basically, the image matching requires slight differences in two successive images for further analysis. Indeed, the ripples are quite dynamic in a river. Therefore, this study suggests a high acquisition frequency (i.e., 20 frames per second) to capture the ripples and to avoid large distinctions in two images. (see lines 141-144)
Lines 141-144:
“Note that the image matching analysis requires slight differences in two successive images. In a river, the ripples are quite dynamic. Therefore, this study used a high acquisition frequency (i.e., 20 fps) to capture the ripples and to avoid large distinctions in two images.”
2) In Figure 11, there is no meaning of equation Y=X (R=0.74). This has to be removed. Further, what is confidence limit for the fitting?
Response 2:
We appreciate the reviewer’s comment and question. The equation Y=X (R=0.74) in Figure 11 (Original version) has been deleted. More details about the linear regression has been provided in the revised manuscript. The averaged surface velocities measured by LSPIV and the flow meter were highly correlated, giving the regression Y=0.916X+0.015, where X and Y represent the results from the flow meter and LSPIV measurements, respectively. The regression with R2 = 0.55 and a P-value ~ 10-8 is statistically significant under a 95% confidence interval. Please see lines 371-375.
Lines 371-375:
“The averaged surface velocities measured by LSPIV and the flow meter were highly correlated, giving the regression Y=0.916X+0.015, where X and Y represent the results from the flow meter and LSPIV measurements, respectively. The regression with R2 = 0.55 and a P-value ~ 10-8 is statistically significant under a 95% confidence interval.”
3) Line 401, how many times the measurements on GCPs are taken. Please indicate instead of several times.
Response 3:
Thanks for the reviewer’s suggestion. In general, the measurements are taken 3 times. The text has been revised. Please refer line 423 to 425 in the revised manuscript.
Lines 423-425:
“In practice, the measurements of GCPs are repeated at least three times and averaged to provide more accurate and less uncertain results for the coordinates of the GCPs.”
4) What is the reason for median of the surface velocities are considered in the comparison and not the mean?
Response 4:
We appreciate the reviewer’s comment. The mean of surface velocities is potentially influenced by the extreme values. The median (or mode) would be a more appropriate way to represent the surface velocities for comparison. Please see lines 483-485.
Lines 486-488:
“Note that the mean of the surface velocities is potentially influenced by extreme values. The median (or mode) would be a more appropriate way to represent the surface velocities for comparison.”
5) What would be the optimum number of GCP's after which the error becomes negligible?
Response 5:
We thank the reviewer’s suggestion. In principle, the accuracy of LSPIV measurement can be improved when a sufficient number of GCPs are used [45]. The number of GCPs depends on how they are used. Generally, at least four to six GCPs should be included. Please see lines 88-91.
Lines 88-91:
“In principle, the accuracy of LSPIV measurement can be improved when a sufficient number of GCPs are used [45]. The number of GCPs depends on how they are used. Generally, at least four to six GCPs should be included.”
References
- Jolley, M. J., Russell, A. J., Quinn, P. F., & Perks, M. T. (2021), Considerations when applying Large-Scale PIV and PTV for determining river flow velocity. Frontiers in Water, 3, 709269. (This reference was provided in the original manuscript)
Finally, we appreciate the reviewer’s feedback again. We hope that the quality of the revision has been improved and the revised manuscript meets the standard of the Journal.

Reviewer 2 Report
The paper has undertaken uncertainty analysis concerning the measurement of GCPs for the use of LSPIV to measure surface velocity in the river. Its topic fits the targeted journal and the focus on the innovative monitoring approach (i.e., the use of LSPIV) could attract readers' interest. However, I found that the paper structures, results presentation, English writing in general are not satisfactory. I will highlight some main problems below.
1) Introduction: The focus of the paper is on the GCPs and its effects on estimated flow velocity and discharge. It should be more clearly demonstrated: how GCP measurements are used in the whole processes? What are the current / existing practices (survey once or multiple times)? More attentions should be one the part of processes that the paper is going to improve on. If you are only use the equations / procedures and then you do not need to detail everything there.
2) Study site and events investigated: There should be more information on the flow conditions on site, such as temporal variability, turbulences, turbidity, seasonality. How representatives are the 4 events studied (normal or high flow)? Why are they chosen? A good monitoring device should perform wells in all weather conditions.
3) Methodology: LSPIV can measure the surface velocity. It is not clear how authors derive the site-specific cross-sectional coefficient (k) and estimate the cross-section velocity. A value was given but no procedures was described. When comparing the results from LSPIV against readings from flowmeter, authors had used the R rather than R^2. The latter is more appropriate as it explain how much variances were explained. By the way, if all R is > 0.95 then a line in the main text will do and no need to tabulate the detailed R values.
As for MCS, please explain how the 3 SE (30mm, 10mm, 3mm) were chosen. Are they the ranges from different types of cameras?
4) Discussion and conclusion: The authors have made comments about how the results agree with existing work. More should be expanded on what are the new and different outcomes. Comments should be made what kind of monitoring is the LSPIV most suitable for, e.g., event-based / long term, moving / fixed point.
5) English writing needs significant improvement. A proof reading / revision by a native speaker is recommended. I have attached annotated version of the paper with my suggestions.
6) Minor issues
Please check the format of section and sub-section headings: some are capitalised and others are not.
Figure and tables:
Figures: There are too many figures. Some should be removed (e.g., Figure 12 and 13, as they present similar information as that of Table 3 and 4) and some could be moved to a supplement document.
Tables: If a table has a column that has same value, then that column should be deleted. The relevant information could be introduced in the main text or footnote for the table.

Author Response
Response to Reviewer 2:
Comments and Suggestions for Authors
The paper has undertaken uncertainty analysis concerning the measurement of GCPs for the use of LSPIV to measure surface velocity in the river. Its topic fits the targeted journal and the focus on the innovative monitoring approach (i.e., the use of LSPIV) could attract readers' interest. However, I found that the paper structures, results presentation, English writing in general are not satisfactory. I will highlight some main problems below.
Response:
We thank the reviewer 2 for the supports of this study. Especially, we appreciate these valuable suggestions for further improvement of the quality of this study.
1) Introduction: The focus of the paper is on the GCPs and its effects on estimated flow velocity and discharge. It should be more clearly demonstrated: how GCP measurements are used in the whole processes? What are the current / existing practices (survey once or multiple times)? More attentions should be on the part of processes that the paper is going to improve on. If you are only use the equations / procedures and then you do not need to detail everything there.
Response 1:
We appreciate the reviewer’s positive comments. The information for the current and existing practices has been added in the revision. The ground control points (GCPs) are required in the LSPIV measurement to obtain orthorectified images. There are two ways to measure GCPs at present: (i) total Station (or electronic distance measuring devices) and (ii) the Global Positioning System (GPS). As the GPS is more expensive, GCPs are usually measured using total stations. In principle, the accuracy of LSPIV measurement can be improved when a sufficient number of GCPs are used [45]. The number of GCPs depends on how they are used. Generally, at least four to six GCPs should be included. Additionally, the surveys for GCPs should be conducted several times and averaged to provide accurate results for the co-ordinates. Please see lines 85-92.
Lines 85-92:
“There are two ways to measure GCPs at present: (i) total stations (or electronic distance measuring devices) and (ii) the Global Positioning System (GPS). As the GPS is more expensive, GCPs are usually measured using total stations. In principle, the accuracy of LSPIV measurement can be improved when a sufficient number of GCPs are used [45]. The number of GCPs depends on how they are used. Generally, at least four to six GCPs should be included. Additionally, the surveys for GCPs should be conducted several times and averaged to obtain accurate results for the coordinates.”
2) Study site and events investigated: There should be more information on the flow conditions on site, such as temporal variability, turbulences, turbidity, seasonality. How representatives are the 4 events studied (normal or high flow)? Why are they chosen? A good monitoring device should perform wells in all weather conditions.
Response 2:
Thanks for the reviewer’s suggestion. More information for the questions above has been provided in the revision. In this study, we considered four normal flow events for the comparison of river surface velocity measured by LSPIV and the flow meter. The main reason for this was to ensure the safety of the surveyors who used the flow meter to measure the river flow for 10 hours during the experimental periods. These benchmark data will be utilized to evaluate the performance/accuracy of LSPIV under its control point set-up. Moreover, a representative event will be used to demonstrate and discuss the uncertainty in LSPIV from the influence of control points. Note that the LSPIV system has shown that it is capable of carrying out long-term continuous monitoring under severe weather conditions [16] although there was no typhoon event in this study. See lines 126-134.
Lines 126-134:
“In this study, we considered four normal flow events for the comparison of river surface velocity measured by LSPIV and the flow meter. The main reason for this was to ensure the safety of the surveyors who used the flow meter to measure the river flow for 10 hours during the experimental periods. These benchmark data will be utilized to evaluate the performance (accuracy) of LSPIV under its control point set-up. Moreover, a representative event will be used to demonstrate and discuss the uncertainty in LSPIV from the influence of control points. Note that the LSPIV system has shown that it is capable of carrying out long-term continuous monitoring under severe weather conditions [16] although there was no typhoon event in this study.”
References
- Huang, W. C., Young, C. C., & Liu, W. C. (2018), Application of an automated discharge imaging system and LSPIV during typhoon events in Taiwan. Water, 10(3), 280. (provided in the original manuscript)
3) Methodology: LSPIV can measure the surface velocity. It is not clear how authors derive the site-specific cross-sectional coefficient (k) and estimate the cross-section velocity. A value was given but no procedures was described. When comparing the results from LSPIV against readings from flowmeter, authors had used the R rather than R^2. The latter is more appropriate as it explain how much variances were explained. By the way, if all R is > 0.95 then a line in the main text will do and no need to tabulate the detailed R values.
As for MCS, please explain how the 3 SE (30mm, 10mm, 3mm) were chosen. Are they the ranges from different types of cameras?
Response 3:
We appreciate the reviewer’s recommendation. Three parts in Methodology section have been improved.
- The cross-sectional mean velocity is estimated using the so-called index velocity method. The index velocity method relates the mean velocity to the surface velocity by a site-specific constant k. In this study, the average surface velocity was measured via LSPIV, and the cross-sectional mean velocity was obtained by the flow meter using the two-point method. The text was modified (see lines 273 to 277).
- The correlation coefficient has been removed from the Table. All the probability density functions have good correlations with R > 0.95. The values of standard errors are listed in Table 3 in the Appendix. A brief description has been added in the main text (see lines 406-408). The information of R squared has been provided in the revised version (see lines 371-375).
- In this study, by three repeated measurements, we obtained a small standard error (SE) about 10 mm for the coordinates of GCPs. Note that continuous efforts have been made to improve the accuracy of experimental instruments over the decades. Here, we considered 3 times and 0.3 times the standard error in the GCP measurement to represent worse and better quality (due to the old- and new-generation instruments or other factors). See lines 299-302.
Lines 273-277: “Further, the cross-sectional mean velocity is estimated using the so-called index velocity method. The index velocity method relates the mean velocity to the surface velocity by a site-specific constant k. In this study, the average surface velocity was measured via LSPIV, and the cross-sectional mean velocity was obtained by the flow meter using the two-point method.”
Lines 371-375: “The averaged surface velocities measured by LSPIV and the flow meter were highly correlated, giving the regression Y=0.916X+0.015, where X and Y represent the results from the flow meter and LSPIV measurements, respectively. The regression with R2 = 0.55 and a P-value ~ 10-8 is statistically significant under a 95% confidence interval.”
Lines 406-408:
“All the probability density functions have good correlations, with R > 0.95. The values of standard errors are listed in Table 3 in the Appendix.”
Lines 299-302: “Note that continuous efforts have been made to improve the accuracy of experimental instruments over the decades. Here, we considered 3 times and 0.3 times the standard error in the GCP measurement to represent worse and better quality (due to the old- and new-generation instruments or other factors).”
4) Discussion and conclusion: The authors have made comments about how the results agree with existing work. More should be expanded on what are the new and different outcomes. Comments should be made what kind of monitoring is the LSPIV most suitable for, e.g., event-based / long term, moving / fixed point.
Response 4:
Thanks for the reviewer’s suggestion. A new paragraph has been added to Conclusion section. This present study further indicates that the uncertainty in LSPIV measurement for the surface velocity of a river can be greatly reduced if the coordinates of the control points can be measured and averaged with three repetitions. The accuracy of LSPIV stream velocity measurement is satisfactory if the standard error for the coordinates of the control points is less than 10 mm. LSPIV systems in a fixed study site have shown they are capable of carrying out long-term continuous monitoring (even under severe weather conditions) [16]. Please see lines 582-588.
Lines 582-588:
" This present study further indicates that the uncertainty in LSPIV measurements of the surface velocity of a river can be greatly reduced if the coordinates of the control points are measured and averaged with three repetitions. The accuracy of LSPIV stream velocity measurement is satisfactory if the standard error for the coordinates of the control points is less than 10 mm. LSPIV systems in a fixed study site have shown that they are capable of carrying out long-term continuous monitoring (even under severe weather conditions) [16]."
5) English writing needs significant improvement. A proof reading / revision by a native speaker is recommended. I have attached annotated version of the paper with my suggestions.
Response 5:
Thanks for the reviewer’s comment. English writing has been modified according to the reviewer’s suggestions in the annotated version of our paper. Also, the revised manuscript has been corrected by the MPDI English editing service.
6) Minor issues
Please check the format of section and sub-section headings: some are capitalized and others are not.
Figure and tables:
Figures: There are too many figures. Some should be removed (e.g., Figure 12 and 13, as they present similar information as that of Table 3 and 4) and some could be moved to a supplement document.
Tables: If a table has a column that has same value, then that column should be deleted. The relevant information could be introduced in the main text or footnote for the table.
Response 6:
Thanks for the reviewer’s suggestions. Minor issues have been corrected. For example, Table 3 and Table 4 have been combined and moved to the Appendix section.
Finally, we appreciate the reviewer’s feedback again. We hope that the quality of the revision has been improved and the revised manuscript meets the standard of the Journal.

Reviewer 3 Report
Comments from the reviewer:
1. Missing heading "4. Streamflow Measurement and Uncertainty Analysis"
2. The gauge station is located at a relatively straight segment of the Creek. One would expect higher surface velocities in the middle, while lower velocities near shore. However, the LSPIV results (Figure 9) show that in general, the surface velocities in the middle are significantly lower than those near both left and right shores. To convince that the LSPIV results are reliable, the authors are required to provide comparisons of the spatial surface velocity distribution across the Creek between LSPIV and flow meter.
3. It would be very useful that a creek cross-section at the gauge station is included in Figure 1.
Author Response
Response to Reviewer 3:
Comments from the reviewer:
Response:
We thank the reviewer 3 for the supports of this study. Especially, we appreciate these valuable suggestions for further improvement of the quality of this study.
- Missing heading "4. Streamflow Measurement and Uncertainty Analysis"
Response 1:
Thank you for the correction. The title (i.e., 4. Results and Discussion) has been added in line 314.
Line 314:
“4. Results and Discussion”
- The gauge station is located at a relatively straight segment of the Creek. One would expect higher surface velocities in the middle, while lower velocities near shore. However, the LSPIV results (Figure 9) show that in general, the surface velocities in the middle are significantly lower than those near both left and right shores. To convince that the LSPIV results are reliable, the authors are required to provide comparisons of the spatial surface velocity distribution across the Creek between LSPIV and flow meter.
Response 2:
This is a good point. We agree with the comment from the reviewer 3. The surface velocities along the cross-section obtained from the LSPIV and flow meter measurements have been shown in Figure 10. The spatial distributions of the river flows are in good agreement, indicating lower velocities in the middle part of the river. As mentioned above, this is attributed to the local topography effects (i.e., higher riverbed or shallower depth due to the shoal). Please see lines 360 to 364.
Lines 360-364:
“The surface velocities along the cross-section obtained from the LSPIV and flow meter measurements are compared in Figure 10. The spatial distributions of the river flows are in good agreement, indicating lower velocities in the middle part of the river. As mentioned above, this is attributed to the local topography effects (i.e., higher riverbed or shallower depth due to the shoal).”
- It would be very useful that a creek cross-section at the gauge station is included in Figure 1.
Response 3:
We thank the reviewer’s comment. A subplot showing the topography of study site (i.e., cross-section elevation) has been added in Figure 1. Please see line 135.
Line 135:
Figure 1. Location map of the study site
Finally, we appreciate the reviewer’s feedback again. We hope that the quality of the revision has been improved and the revised manuscript meets the standard of the Journal.

Round 2
Reviewer 2 Report
Authors have addressed all my concerns and the paper has been significantly improved and I am happy to support the publication of the paper in current form.